Fisetin may protect early porcine embryos from oxidative stress by down-regulating GRP78 levels

Yuan Xiu-Wen 1
Guo Hao 1 2
Wang ChaoRui 1
Ji HeWei 1
Xu YongNan 1
Yao Xue Rui 1
Wang Lin 3
Cao QiLong 3
Kim Nam-Hyung 1
Li Ying-Hua yhli@wyu.edu.cn 1
1 Guangdong Provincial Key Laboratory of Large Animal Models for Biomedicine, South China Institute of Large Animal Models for Biomedicine, School of Pharmacy and Food Engineering, Wuyi University , Jiangmen , China
2 College of Light Industry and Chemical Engineering, GuangDong University of Technology , Guangzhou , Guangdong , China
3 Qingdao Branch, Qingdao Haier Biotechnology , Qingdao , Shandong , China
Sun Shao-Chen
Electronic publication date: 2025 Mar 28
Publication date: 2025
Volume: 13
Electronic Location ID: e19198
Received 2024 Dec 12; Accepted 2025 Mar 3
Copyright: ©2025 Yuan et al.
Copyright year: 2025
Copyright holder: Yuan et al.
License: This is an open access article distributed under the terms of the Creative Commons Attribution License, which permits unrestricted use, distribution, reproduction and adaptation in any medium and for any purpose provided that it is properly attributed. For attribution, the original author(s), title, publication source (PeerJ) and either DOI or URL of the article must be cited.
License URL: https://creativecommons.org/licenses/by/4.0/

Keywords: Fisetin, GRP78, Oxidative stress, Embryo quality

Funding: The Science and Technology Planning Project of the Guang-dong Provincial Department of Science and Technology 2021B1212040016 Special project in key areas of biomedicine and health of Guangdong Provincial Department of Education 2021ZDZX2046 The CAS KeyLaboratory of Regenerative Biology, Guangdong Provincial Key Laboratory of Stem Cell and Regenerative Medicine, Guangzhou lnstitutes of Biomedicine and Health, Chinese Academy of Sciences KLRB202204 The “Chunhui Plan” cooperative scientific research project of Guangdong Province Department of Education 202202131 This study was supported by the Science and Technology Planning Project of the Guang-dong Provincial Department of Science and Technology (Project No.: 2021B1212040016), special project in key areas of biomedicine and health of Guangdong Provincial Department of Education (Project No.: 2021ZDZX2046), the CAS KeyLaboratory of Regenerative Biology, Guangdong Provincial Key Laboratory of Stem Cell and Regenerative Medicine, Guangzhou lnstitutes of Biomedicine and Health, Chinese Academy of Sciences (Project No.: KLRB202204) and the “Chunhui Plan” cooperative scientific research project of Guangdong Province Department of Education (Project No.: 202202131). The funders had no role in study design, data collection and analysis, decision to publish, or preparation of the manuscript.

==============================
Fisetin is a natural flavonol with a variety of biological activities, including anti-inflammatory and antitumor activities. However, the effect of fisetin on mammalian oocyte and embryo development is unknown, so in this study, we used porcine oocytes as an experimental model, and added optimal concentrations of fisetin to the in vitro culture medium after parthenogenetic activated to investigate the effect of fisetin on porcine embryo development. It was found that 0.1 µM fisetin significantly increased the cleavage rate and blastocyst formation rate, and the quality of blastocysts was also improved. Staining results showed that the levels of reactive oxygen species (ROS), autophagy, endoplasmic reticulum stress and apoptosis were significantly reduced, while glutathione levels and mitochondrial function were significantly increased in the 0.1 µM fisetin-treated group of early porcine embryos compared with the control group. Meanwhile, fisetin decreased the expression level of the endoplasmic reticulum stress marker protein GRP78 (0.71 ± 0.19). In addition, fisetin decreased the expression of genes related to pro-apoptosis, autophagy and endoplasmic reticulum stress and increased the expression of genes related to antioxidant, pluripotency and mitochondrial. According to our results, fisetin promotes early embryonic development in porcine, and this effect may be realized by down-regulating the expression level of GRP78.

Introduction

Embryo production in vitro is one of the most important techniques to improve animal genetics, and it has been widely applied in agriculture, assisted reproductive technology, and study on the early stages of animal development (Ealy, Wooldridge & McCoski, 2019; Wrenzycki, 2018). It is considered an important tool for improving genetic gains through the maternal line, so the selection of female egg quality is no less intense than the high standard of sperm in artificial insemination (Manik, Singla & Palta, 2003; Perez et al., 2019). Due to the similarity between porcine and human epigenetics, porcine oocytes and embryos are considered a practical model for investigating how diet affects human ovaries (Shi & Sirard, 2022; Sjunnesson, 2020; Teng et al., 2024). However, despite the continuous optimization of in vitro maturation (IVM) and development media formulations, it is difficult to achieve ideal embryo culture conditions with in vitro culture (IVC) technology at present. Both embryo quality and subsequent development remain quite low, particularly in systems of in vitro production for porcine embryos (De Souza Fabjan et al., 2014; Chen et al., 2021b). Therefore, continuously developing and optimizing these cultural systems are necessary to raise the standard of porcine early embryo development.

Among the many fruits and vegetables are fisetin (3,30,40,7-tetrahydroxyflavone; FIS), a biologically active flavonoid that can be found in cucumbers, persimmons, grapes, strawberries, apples, onions (Guo et al., 2024; Mahoney et al., 2023; Harrison et al., 2023), with chemopreventive potential (Kashyap et al., 2019). FIS reduces cellular senescence in human endothelial cell cultures (Mahoney et al., 2023; Hassan et al., 2022), and using aged mice as a research model, FIS delayed the aging of oocytes after ovulation (Xing et al., 2023). FIS’s antioxidant properties protect nerve cells from inflammation and apoptosis degeneration (Ahmad et al., 2019; Fernanda Arias-Santé et al., 2024). In addition, due to its multiple biological activities, FIS is regarded as a health-promoting substance, and there are currently a number of dietary supplements available that include FIS (Khan et al., 2013). However, to date, the effects of FIS on early embryonic development in pigs and the possible mechanisms of action are unknown.

The primary location for cellular biological oxidation and energy conversion is mitochondria and produce ATP for cellular activities, accompanied by the regulation of amounts of reactive oxygen species (ROS) (Zorov, Juhaszova & Sollott, 2014; Rodríguez-Varela & Labarta, 2020). The excessive accumulation of ROS in mitochondria induces DNA damage, cellular oxidative stress, and mitochondrial dysfunction (Li, Zheng & Ding, 2022). An important factor in oocyte apoptosis is the mitochondrial pathway triggered by oxidative stress, which possibly results in fertilization failure and early embryo development arrest (Lee, Adhikari & Carroll, 2022; Zhou et al., 2022; Stockmann-Juvala & Savolainen, 2008; Tiwari et al., 2015; Abate et al., 2020).

The endoplasmic reticulum (ER) is a multifunctional organelle responsible for cellular homeostasis, protein synthesis, folding, and secretion (So, 2018). Glucose-regulated protein 78 (GRP78) is a member of the heat shock protein family, which is required by the ER in the stress response (Li & Lee, 2006) and is mainly involved in protein folding and degradation of misfolded proteins (Lei et al., 2024). GRP78 is involved in mediating cell cycle protein levels with normal mitochondrial function during porcine embryonic development (Heo et al., 2023). Studies have shown that when porcine embryos are exposed to external stimuli that trigger OS and subsequent ERS, GRP78 levels are elevated and early embryonic developmental potential is reduced (Liu et al., 2023).

Endoplasmic reticulum stress is a defense mechanism of cells in response to external stimuli, such as intraluminal misfolding and unfolded protein aggregation, and the dysregulation of calcium homeostasis. External stimuli (e.g., ROS) can disrupt the homeostasis of the ER, which can hinder the maturation of female mammalian oocytes and impair early embryonic development (Lin et al., 2019). The aim of this study was to elucidate the effects of FIS on mitochondria and endoplasmic reticulum during porcine oocyte development and to attempt to probe deeper.

Material and methods

Reagents

Unless otherwise specified, the study’s whole supply of chemicals and reagents came from Sigma-Aldrich (St. Louis, MO, USA). And both pituitary follicle-stimulating hormone (FSH; 110254629-100IU) and luteinizing hormone (LH; 110254634-200IU) are derived from Ningbo Second Hormone Factory (Zhejiang, China). Fisetin (HY-N0182; MedChemExpress, Shanghai, China), the natural active substance, had a purity of 98.91%. It was dissolved in dimethyl sulfoxide (DMSO; D2650-100ml) to form a one mM stock solution stored at −80 °C. DMSO final percentage not to exceed 1%.

Ethics statement

The experimental methods used in the study, including animal experiments, were carried out in accordance with the relevant guidelines and approved by the Animal Care and Use Agency Committee of Wuyi University.

Oocyte harvesting

The ovaries from prepubertal gilts used in this study were provided by local slaughterhouses. The acquired porcine ovaries were immediately placed into a 37 °C Thermos flask filled with 0.9% sterile saline. They were transported to the laboratory within 2 h. The ovaries were washed 3–4 times with 37 °C normal saline. Follicular fluid and cumulus–oocyte complexes (COCs) from 3–8 mm diameter follicles were aspirated with a 10 mL sterile syringe attached to an 18 gauge needle. The COCs were washed three times in Tyrode’s lactate HEPES medium, transfer approximately 100 COCs into four-well plates, each well containing 500 µL IVM and covered with an equal volume of mineral oil. IVM medium concisting of medium 199 supplemented with 0.1 mg/ml sodium pyruvate, 0.01 ug/ml epidermal growth factor, 10 IU/ml pituitary follicle stimulating hormone and 10 IU/ml luteinizing hormone. Incubate in a 38.5 °C incubator for 44–46 h with 100% relative humidity and 5% CO2 content. There was no need to change the IVM medium during IVM.

Parthenogenetic activation

After 46 h of IVM, the COCs were transferred to a 30 mm dish containing 0.01% hyaluronidase solution, remove the cumulus cells to obtain bare oocytes. The activation fluid, which contained 3 × 10−1 mol mannitol, 5 × 10−4 mol HEPES, 5 × 10−5 mol CaCl2•2H2O, 1 × 10−4 mol MgSO4•7H2O, polyvinyl alcohol and bovine serum albumin (BSA) 0.1 mg/ml and one mg/mL, respectively, was added to the fully nude oocytes. It was then activated with 120 V DC pulsed electricity (CFB16-HB; BioRad; Hercules, CA, USA), a process that was cycled twice for 60 µs each. Next, our parthenogenetic-activated oocytes were placed in IVC medium containing four mg/mL BSA and 7.5 µg/mL of cytochalasin B and moved for four hours to a 38.5 °C, 5% CO2 incubator to prevent the second polar body from being excluded. After 4 h, wash the oocytes 4–5 times with IVC medium (bicarbonate buffer containing four mg BSA per 10 ml). Subsequently, approximately 40 oocytes were cultured in 50 µL of IVC medium with or without FIS for 7 days. During this time, no change of medium was required. The cleavage and blstocyst formation rates were recorded at days 2 and days 7, respectively.

ROS levels and glutathione levels measurement

After 48 h of in vitro culture, at least 15 four-cell stage embryos were selected and placed in dye H2DCFDA (Beyotime, Shanghai, China) and CMF2HC (Beyotime, Shanghai, China) was incubated in phosphate-buffered saline and polyvinyl alcohol (PBS/PVA) buffer for 30 min (during which the temperature should be 37.5 °C in a dark environment) to measure intracellular ROS and GSH levels. After 3 washes with PBS/PVA at the end, green and blue fluorescence were respectively imaged using an inverted fluorescence microscope (Ti2eU; Nikon, Tokyo, Japan), and the fluorescence intensity was analyzed with ImageJ version 8.0.2 software (NIH, Bethesda, MD, USA).

Mitochondrial function evaluation

To detect the distribution of mitochondria, oocytes at four cell development stages were incubated at 38.5 °C, 5% CO2 and in the dark for 1 h in PBS/PVA buffer containing 10 µmol/L 5,5′,6,6′-tetrachloro-1,1′,3,3′-tetraethylbenzi-midazolylcarbocyanineiodide iodide (JC-1, Beyotime, Shanghai, China) fluorescent probe to detect mitochondrial membrane potential. After that, five PBS/PVA washes were carried out, an inverted fluorescence microscope was used to identify the red-green fluorescence signal in oocytes, and ImageJ version 8.0.2 software was used to assess the red-green fluorescence intensity of JC-1. The average mitochondrial membrane potential was assessed by calculating the ratio.

Immunofluorescence staining

After washing the blastocysts with PBS/PVA on the seventh day, they were fixed in 4% paraformaldehyde for 30 min. Afterward, the blastocysts were permeabilized for 30 min using 0.5% Triton X-100 and then blocked for one hour in PBS/PVA containing 1% BSA. Subsequently, the blastocysts were treated with rabbit anti-LC3B antibody (1:200; ab48394; Abcam, Cambridge, UK) and rabbit anti-GRP78 antibody (1:300; ab21685; Abcam, Cambridge, UK) overnight at 4 °C. After four washes in PBS, incubate for 1 h at room temperature using goat anti-rabbit IgG antibodies (Alexa Fluor® 488; 1:500; ab150077; Abcam, Cambridge, UK). Then stain with Hoechst 33342 for 10 min to mark the location of the nuclei. Finally, wash blastocysts with PBS/PVA buffer and after four replicates, transfer them to glass slides and deal with fluorescence intensity.

Detection of total cell count and blastocyst proliferation levels

BeyoClick EdU-647 Assay Kit (Beyotime, Shanghai, China) was used to detect blastocyst cell proliferation. On the 6th day after parthenogenetic activation of oocytes, add 10 µM EdU to in vitro culture medium and incubate at 38.5 °C, 5% CO2 and in the dark for 8 h. Next, the blastocysts are incubated with BeyoClick additive solution for 1 h in the dark and the nuclei are located with Hoechst 33342. At the end of staining, the blastocysts are fixed on a glass slide. The number of positive cells was detected using an inverted fluorescence microscope to evaluate the proliferation ability of blastocyst cells.

dUTP notched end marker (TUNEL) staining

The TUNEL assay kit (Roche Diagnostics, Indianapolis, IN, USA) is used to detect the level of apoptosis at blastocyst stage. On the seventh day, blastocysts were washed with PBS/PVA and then treated with 3.7% paraformaldehyde and 0.1% Triton X-100 sequentially. Next, embryonic apoptotic nuclei are labeled using the TUNEL kit. Then, Hoechst 33342 (Beyotime, Shanghai, China)) 10 µg/mL was incubated for 8 min in darkness, after four rinses in PBS/PVA, the slides with fixed with antifluorescence attenuation sealant (Boster, Hubei, China). All of the above steps were performed at room temperature. The number of apoptotic nuclei in a blastocyst was used to measure its level of apoptosis.

RNA extraction and quantitative real-time PCR

On the seventh day after parthenogenetic activation, 20 blastocysts of the same quality were collected from the control group and the treatment group. The Dynabeads mRNA DIRECT kit (Invitrogen, Waltham, MA, USA) was used to extract embryonic total mRNA as instructed by the manufacturer. Then, the extracted mRNA was reverse-transcribed with the DynaBeads mRNA DIRECT kit (Life Technologies, Carlsbad, CA, USA) to obtain cDNA. Using the KAPA SYBR FAST qPCR Master Mix (2X) Kit, the levels of gene expression were determined. KAPA SYBR Green 10 µL, cDNA 0.2 µL, forward and reverse primers 0.4 µL each, ddH2O nine µL, the above substances constitute the whole sample system. The PCR cycle conditions are as follows: pre-denaturalization at 95 °C for 3 min, denaturalization for 3 s at 95 °C, 30 s of annealing at 60 °C, 20 s prolonged at 72 °C, and 40 cycles in a 3-step method. The target genes were LC3B, ATG5, BECLIN, P62, OCT4, NANOG, ESRRB, SOX2, UCHL1, SOD2, SIRT1, CAS3, CAS8, BAX, BCL2, eIF2α, GRP78, CHOP, ATF4, ATF6, uXBP1, sXBP1, Nrf2, PGC1α, and TFAM. The 2−ΔΔCt technique was utilized to determine the relative mRNA expression of the target genes, with GAPDH serving as an internal control. The primer sequences of the genes used in this study are presented in Table 1.

Table 1 Primer sequences used for quantitative real-time PCR.

Genes	Primer sequences (5′–3′)	Base	
GAPDH	F: GTCGGTTGTGGATCTGACCT	20	
R: TTGACGAAGTGGTCGTTGAG	20	
SOX2	F: GAACAGCCCAGACCGAGTTA	21	
R: ATCTTGGGGTTCTCTTGGGC	22	
NANOG	F: CAGTGATTTGGAGGCCGTCT	20	
R: TCCATGATTTGCTGCTGGGT	20	
ESRRB	F: CCGGACAAACTCTACGCCAT	20	
R: GAGAAGCCTGGGATGTGCTT	20	
OCT4	F: CCTATGACTTCTGCGGAGGGA	21	
R: TTTGATGTCCTGGGACTCCTCG	22	
CAS3	F: TTGAGACGGACAGTGGGACT	20	
R: CCGTCCTTTGAATTTCGCCAG	21	
CAS8	F: GCCTCGGGGATACTGTTTGA	20	
R: CGCTGCATCCAAGTCTGTTC	20	
BAX	F: GGACTTCCTTCGAGATCGGC	20	
R: GCGTCCCAAAGTAGGAGAGG	20	
BCL2	F: GGATAACGGAGGCTGGGATG	20	
R: TTATGGCCCAGATAGGCACC	20	
UCHL	F: ACTTTGGATTCGCTCGGTAC	20	
R: CGCTTATCTGCAGACCCCAA	20	
SOD2	F: CTGCAAGGAACAACAGGTCT	20	
R: CTGCAAGGAACAACAGGTCT	20	
SIRT1	F: ACAGGTTGCAGGAATCCAGAG	21	
R: TAGGACATCGAGGAACCACCT	21	
Nrf2	F: AGCGGATTGCTCGTAGACAG	20	
R: TTCAGTCGCTTCACGTCGG	19	
PGC1α	F: TTCCGTATCACCACCCAAAT	20	
R: ATCTACTGCCTGGGGACCTT	20	
TFAM	F: TCCGTTCAGTTTTGCGTATG	20	
R: TTGTACACCTGCCAGTCTGC	20	
LC3B	F: TTCAAACAGCGCCGAACCTT	20	
R: TTTGGTAGGATGCTGCTCTCG	21	
ATG5	F: TTGCAGTGGCTGAGTGAACA	20	
R: TCAATCTGTTGGTTGCGGGA	20	
BECLIN	F: CATGAAGATGACAGCGAACAGC	22	
R: AGATTTTCCGCCACTATCTTCCG	23	
P62	F: AAGAACGTAGGGGAGAGTGTG	21	
R: TTCCCTCCATGTTCCACGTC	20	
eIF2α	F: ACAACCACCCTGGAGAGAACA	21	
R: TATCTGTAACCACTTTGGGCTCC	23	
GRP78	F: GCTCTACTCGCATCCCCAAAG	21	
R: TACACCAGCCTGAACAGCAG	20	
CHOP	F: CCCCTGGAAATGAGGAGGAG	20	
R: CTCTGGGAGGTGTGTGTGAC	20	
ATF4	F: AGTCCTTTTCTGCGAGTGGG	20	
R: CTGCTGCCTCTAATACGCCA	20	
ATF6	F: TACTTCCAGCAGCACCCAAG	20	
R: GCACCACCGTCTGACCTTTA	20	
uXBP1	F: CATGGATTCTGACGGTGTTG	20	
R: GTCTGGGGAAGGACATCTGA	20	
sXBP1	F: GGAGTTAAGACAGCGCTTGG	20	
R: GAGATGTTCTGGAGGGGTGA	20	
Notes.

F Forward primer

R Reverse primer

Western blotting

A total of 30 blastocysts at grade 1 from each group were fully lysed with RIPA lysate containing 1% one mM PMSF. Add 5X protein loading buffer, mix well and cook in a 100 °C water bath for 10 min. Denatured proteins were separated by SDS-PAGE using a 4%–20% (w/v) gel and transferred to a polyvinylidene fluoride (PVDF) membrane (Millipore, Burlington, MA, USA). For 1 h at room temperature, membranes were blocked in 5% (w/v) BSA in Tris-buffered saline (TBS) with 0.1% (w/w) Tween 20 (TBST). The PVDF membranes were transferred to rabbit anti-GRP78 antibody (1:1000; ab21685; ab21685; Abcam, Cambridge, UK) and Anti-β-actin (1:1000; 4970; Cell Signaling Technology, Danvers, MA, USA) primary antibody incubation solution at the end of the sealing process, and kept at a constant temperature of 4 °C and overnight. After three washes (8 min each) with TBST, PVDF membranes were incubated with Anti-rabbit IgG, HRP-linked antibody (1:10,000; Cell Signaling Technology, Danvers, MA, USA) for 2 h at room temperature. Afterward, the membranes were washed three times (8 min each) in TBST, then exposed to an enhanced chemiluminescent reagent (Tanon, Shanghai, China), and the protein bands were displayed by the detection system (Azure Biosystems, Dublin, CA, USA). The grayscale corresponding to each band is analyzed by ImageJ software (NIH, Bethesda, MD, USA).

Statistical analysis

All experiments performed in this study were repeated at least three times, with a sample size of no less than 10 each time. The mean ± standard deviation (SD) are used to calculate the results. The N in the image represents the sample size of each experiment, and R represents the number of experimental replicates. When comparing two groups, the t-test was employed, and when comparing three or more groups, the ANOVA (Tukey–Kramer) was utilized. The statistical analysis was performed using SPSS version 22.0 software (IBM Corp, Armonk, NY, USA), and use GraphPad Prism software to make histograms. When p < 0.05, p < 0.01 and p < 0.001 are respectively represented as *, **, *** in the histogram, NS stands for data not significantly different.

Results

Effect of FIS supplementation on porcine embryonic development

We chose to add 0, 0.01, 0.1, and one µM of FIS to IVC medium for formal experiments, and the control group was the zero µM. The cleavage rate was measured 48 h after parthenogenetic activation, and the cleavage rate between the control group and the FIS treatment group at varied concentrations did not differ significantly (Fig. 1B). The quality of blastocysts was significantly better than that of the control group after the addition of FIS (Fig. 1A). The blastocyst formation rate on day 7 was 36.70 ± 7.07% (0 µM), 39.09 ± 7.29% (0.01 µM), 48.56 ± 8.15% (0.1 µM), and 30.35 ± 7.77% (one µM) (Fig. 1C). Among them, the blastocyst formation rate was significantly increased in the 0.1 µM FIS treatment group. Therefore, subsequent experiments were carried out at this concentration.

Figure 1 Effect of different FIS concentrations (0, 0.01, 0.1, one µM) on porcine embryo production in vitro.

(A) On the seventh day, the groups treated with FIS (0 µM, 0.01, 0.1, and one µM) all showed signs of embryo development. Scale bar = 100 µm. (B) Effects of different concentrations of FIS on cleavage during porcine embryonic development (R = 3). (C) Blastocyst formation rate at different concentrations of FIS (N = 239, R = 3). *P < 0.05, ***P < 0.001.

FIS treatment improves the proliferative capacity of blastocyst cells

After identifying the positive effects of FIS on oocyte development, we assessed how FIS affected the growth of embryonic cells. The FIS group had a considerably larger proportion of EdU-positive cells than the control group (0.34 ± 0.12 vs. 0.27 ± 0.13; Figs. 2A and 2B). After staining the blastocyst cell nuclei, the results showed that the total number of nuclei in the blastocysts were significantly higher than that in the control group after 0.1 µM FIS treatment (49.65 ± 12.51 vs. 42.41 ± 12.32; Figs. 2C and 2D). In addition, the expression levels of genes OCT4 (1.39 ± 0.20), NANOG (2.21 ± 0.05), ESRRB (1.14 ± 0.07), and SOX2 (2.42 ± 0.52) encoding pluripotency-related factors were significantly higher than those in the control group (Fig. 2E). The data above suggest that FIS can promote cell proliferation in the early stages of embryonic development.

Figure 2 Effect of FIS on the proliferation of blastocysts.

(A) Representative EdU staining of blastocysts. Scale bar = 50 µm. (B) Percentages of EdU-positive nuclei in the control group and the FIS-treated (0.1 µM) group on day 7 (N = 48 vs. N = 38; R = 3, p < 0.01). (C) Staining image of the nuclei within the blastocyst. Scale bar = 50 µm. (D) Histogram of the total number of nuclei within the blastocyst on day 7 for both the FIS-treated group (0.1 µM) and the control group. (E) Promoting the mRNA expression level of genes related to cell proliferation. *P < 0.05, **P < 0.01, ***P < 0.001.

FIS inhibits apoptosis during embryonic development

The apoptosis rate in blastocysts was detected by in situ end labeling. The results showed that the level of apoptosis was significantly reduced after 0.1 µM FIS treatment compared with the control group(0.04 ± 0.02 vs. 0.08 ± 0.04; Figs. 3A and 3B). Quantitative real-time PCR results showed the same trend; compared with the control group, the anti-apoptotic gene BCL2 (1.23 ± 0.07) was significantly upregulated, and BAX (0.45 ± 0.27), CAS8 (0.29 ± 0.01) and BAX/BCL2 (0.36 ± 0.20)were significantly reduced in the FIS group (Fig. 3C).

Figure 3 Effect of FIS on the apoptosis of blastocysts.

(A) Representative photos of blastocysts colored with TUNEL test. Scale bar = 50 µm. (B) Blastocyst apoptosis of control (N = 33) and 0.1 µM FIS (N = 35) groups at day 7 (R = 3, p < 0.001). (C) Impact of FIS on gene expression levels linked to cell death. *P < 0.05, **P < 0.01, ***P < 0.001.

Figure 4 Impact of FIS on GSH and ROS expression levels in embryos.

(A) GSH standardized graph for four-cell period. Scale bar = 100 µm. (B) Relative levels of GSH in embryos with (N = 77) or without (N = 64) FIS treatment (R = 3, p < 0.001). (C) ROS Staining Standard Chart. Scale bar = 100 µm. (D) ROS fluorescence intensity in embryos treated with (N = 41) or without (N = 41) fluorescence induction system (R = 3, p < 0.001). (E) FIS treatment’s effects on genes linked to intracellular oxidative stress. *P < 0.05, **P < 0.01, ***P < 0.001.

FIS treatment enhanced the early antioxidant activity in porcine embryos

To investigate the fundamental processes of FIS involvement in promoting embryonic development, we measured the levels of 4-cell phase GSH and ROS during oocyte development (Figs. 4A and 4C). FIS can significantly increase the GSH level in embryo of 4-cell stage (p < 0.001; Fig. 4B). In contrast, Early embryos in the FIS treatment group had far lower levels of ROS than those in the control group (p < 0.001; Fig. 4D). In embryos treated with FIS, the UCHL1, SOD2, and SIRT1 genes were elevated (2.41 ± 0.11, 1.08 ± 0.03, 1.40 ± 0.40, respectively; Fig. 4E).

Adding FIS to the medium improves mitochondrial activity

The mitochondrial membrane potential is crucial for early embryonic development in porcine and an essential indicator of mitochondrial function. The 4-cell phase mitochondrial function was tested (Fig. 5A). The results showed that the red/green fluorescence intensity after FIS treatment was significantly higher than that of the control group (p < 0.001; Fig. 5B). Furthermore, Nrf2, PGC1α, and TFAM (genes linked to mitochondrial biogenesis) had correspondingly high levels in embryos (2.22 ± 0.21, 1.66 ± 0.12, 1.56  ± 0.29; Fig. 5C).

Figure 5 Impact of FIS on early embryonic mitochondrial activity.

(A) A typical image of 4-cell embryos stained with JC-1. Scale bar = 100 µM. (B) Comparative red/green JC-1 levels in the FIS treatment and the control (N = 57 vs. N = 48; R = 3). (C) Variations in mitochondrial function-related gene expression levels following the addition of FIS. *P < 0.05, ***P < 0.001.

FIS reduces the level of autophagy during porcine embryonic development

Excess ROS induces high levels of intracellular autophagy during the in vitro culture of oocytes. We measured the amount of LC3B-positive cells, which are highly correlated with autophagy levels, in order to evaluate the impact of FIS on autophagy (Fig. 6A). The level of autophagy (0.69 ± 0.18, p < 0.001; Fig. 6B) in the FIS treatment group was significantly reduced. Moreover, autophagy genes (LC3B, ATG5, P62) of the FIS-treated embryos were reduced (0.30 ± 0.21, 0.61 ± 0.12, 0.65 ±0.17; Fig. 6C).

Figure 6 Influence of FIS on early embryo autophagy.

(A) Typical image of LC3B stained blastocysts. Scale bar = 100 µm. (B) Relative LC3B fluorescence intensity levels in FIS-treated blastocysts compared to untreated (N = 40 vs. N = 35, R = 3, p < 0.001). (C) Expression levels of autophagy genes after FIS treatment (p < 0.05). *P < 0.05, ***P < 0.001.

Figure 7 Effect of FIS on ERS levels in early embryos.

(A) An illustration of blastocysts stained with GRP78. Scale bar = 50 µm. (B) Relative levels of GRP78 fluorescence intensity in blastocysts with (N = 60) or without (N = 43) FIS treatment (R = 3, p < 0.001). (C) The protein content of GRP78 in blastocysts on day 7 was shown as a band gray value (GRP78/β-actin). (D) Relative expression of GRP78 protein. (E) Effect FIS on ERS gene expression levels in blastocysts. *P < 0.05, **P < 0.01, ***P < 0.001.

FIS protect early porcine embryos from ERS caused by external stimuli

The ER and mitochondria are interrelated during mammalian growth and development. Therefore, we detected the expression of the ERS marker GRP78. Compared to the control group, the GRP78 fluorescence level in the FIS group was considerably lower (0.92 ± 0.07; Figs. 7A and 7B). In order to confirm the effect of FIS on the ER of early porcine embryos further, the protein expression levels of GRP78 was detected (Fig. 7C). The results were consistent with the previous results, with a decrease in GRP78 protein expression (1.00 vs 0.71 ± 0.19; Fig. 7D). The ERS-related gene expression in the FIS treatment group (eIF2α, GRP78, CHOP, ATF6, ATF4, uXBP1, sXBP1) was decreased (0.74 ± 0.15, 0.72 ± 0.09, 0.42 ± 0.14, 0.67 ±0.04, 0.14 ± 0.07, 0.96 ± 0.03, 0.78 ± 0.11; Fig. 7E). The results showed that FIS could reduce ERS in early porcine embryos and promote their developmental potential.

Discussion

In this study, the blastocyst formation rate was dramatically boosted by supplementing with 0.1 µM FIS. The total number of cells and the capacity for cell proliferation in the FIS treatment group were significantly higher than in the control group, which is consistent with the action of more antioxidants (Wang et al., 2022b; Liu et al., 2022; Wang et al., 2023). The pluripotency genes (ESRRB, SOX2, OCT4, NANOG) expression was significantly increased. In particular, the high expression of NANOG in the FIS treatment group will help the embryo to improve the expression of pluripotency genes and promote embryonic development during development (Navarro et al., 2012). Conversely, it has also been shown that high concentrations of FIS inhibit the subsequent development ability of embryos. This is consistent with a previous study that FIS inhibits the proliferative ability of cancer cells and induces apoptosis at high concentrations (Afroze et al., 2022). Therefore, it is of great significance to further explore the specific mechanism of the positive effect of FIS on embryonic development.

The ATP produced by mitochondria is used for the normal development of porcine embryos. This process is accompanied by the production of by-products, such as ROS, which are present as free radicals (O2−), hydrogen peroxide (H2O2), hydroxyl radicals (•OH), and organic peroxides (Matés et al., 2012). Zygotic genome activation (ZGA) is a central event in early animal embryonic development, marking the onset of the maternal-syntenic transition (Zhai et al., 2022). During pig early embryonic development, ZGA occurs at the 4-cell stage. This stage is critical for the further development of the embryo. Due to the antioxidant capacity of FIS in some studies, we measured the antioxidant index in 4-cell embryos. We found that FIS treatment significantly decreased ROS accumulation in the 4-cell stage while simultaneously increasing GSH content. It may be because FIS can improve oocytes’ ability to cope with oxidative stress through the SIRT1 pathway (Xing et al., 2023). Intracellular ROS and GSH levels are closely related. The increase in antioxidant capacity in embryos may also be because FIS can induce high levels of GSH, enhance the capacity to scavenge excess free radicals of oxygen and maintainoxidation–reduction balance (Yu et al., 2023). The antioxidant activity of FIS may affect the early embryo development of porcine. However, further research into the precise mechanism is still necessary.

Mitochondria are the main energy supply stations in the cytoplasm and are necessary for early embryo development (Tarazona et al., 2006). The mitochondrial membrane potential is associated with the early embryo developmental potential (Romek et al., 2011). In this study, the level of matrix protease in embryos was significantly raised by FIS treatment. FIS was found to dramatically up-regulate the mRNA levels of TFAM, Nrf2, and PGC1α, indicating its potential to improve mitochondrial function during embryonic development. It is speculated that FIS may stimulate the SIRT1 pathway and upregulate SIRT1 expression, positively affecting mitochondrial biogenesis and gene expression.

The external environment stimulates embryos, and then ERS occurs, negatively affects the maturation and subsequent development of porcine oocytes, and causes a certain degree of damage to the mammalian reproductive system (Ridlo et al., 2021; Capatina et al., 2021; Chen et al., 2021a). Based on the experimentally proven results that FIS treatment protect embryos from oxidative stress and improves mitochondrial function, antioxidant capacity and cell proliferation capacity, we hypothesized that FIS may improve the quality of porcine embryos by reducing intracellular ERS. The results showed that FIS treatment decreased the ERS marker GRP78 and ERS related genes (eIF2α, GRP78, CHOP, etc.), and the expression intensity of protein (GRP78) in blastocysts was also decreased. In turn, the unfolded protein response (UPR) is activated by the PERK/eIF2α/ATF4, IRE1/XBP1, and ATF6 pathways for downstream transcriptional programs, thereby enhancing the ability of proteins to fold in the ER (Yang et al., 2016; Basar et al., 2014). High levels of UPR signaling pathways are associated with endoplasmic reticulum stress genes such as GRP78 and CHOP, and it will induce poor oocyte quality and embryonic dysplasia are closely related (Liu et al., 2021). Consequently, our research indicates that FIS therapy may lessen ERS.

It is well recognized that ERS and oxidative stress are strongly associated with apoptosis levels. In order to explore the potential protective mechanism of FIS on embryonic development, we measured the level of apoptosis (TUNEL). FIS treatment not only reduced the incidence of apoptosis, but also increased GSH levels and reduced ROS accumulation. Additionally, oxidative stress initiates the apoptotic pathway and increases the expression of CAS3 and CAS8 (Yu et al., 2017; Wang et al., 2022a; Wang et al., 2019). The quantitative real-time PCR results also confirmed this view, and FIS supplement significantly upregulated the anti-apoptotic gene BCL2 and downregulated the expression of the pro-apoptotic genes BAX, CAS3, and CAS8. These results suggest that FIS can inhibit oxidative stress and apoptosis caused by endoplasmic reticulum stress.

Autophagy is the process in which eukaryotes use lysosomes to degrade their cytoplasmic proteins and damaged organelles. Various physiological and pathological conditions induce autophagy, and autophagy-related genes have a critical role in facilitating the regulation of well-coordinated autophagy (Ohsumi, 2001; Parzych & Klionsky, 2014). When a cell initiates an autophagy program, it generates ROS through oxidative stress, which may lead to cell death (Chen et al., 2008). Autophagy is necessary for oocyte development, and at the same time, early embryonic development and oocyte maturation are hampered by excessive autophagy (Zhang et al., 2022). In this study, the levels of LC3B in the FIS treatment group were significantly reduced compared with the control group, and the mRNA levels of ATG5, LC3B, and p62 were also significantly decreased. These findings imply that porcine early embryos treated with FIS may offer defense against harm brought on by over-autophagy. Finally, since the in vitro culture environment of oocytes does not fully mimic the in vivo microenvironment, in vivo experiments using different species of animals as research models are still needed to further validate the effects of FIS on early embryonic development in the future.

Conclusions

In conclusion, supplementation of 0.1 µM fisetin in the in vitro culture medium of porcine embryos can improve the antioxidant capacity and mitochondrial function of porcine embryos through the GRP78 pathway, as well as reduce apoptosis, autophagy and endoplasmic reticulum stress levels within the early porcine embryo, thereby improving the quality of early porcine embryos. However, the potential efficacy of FIS to improve embryo quality needs to be further investigated through in vivo experiments in animals, and the establishment of both optimal dosages and treatment regimens will be necessary.

Supplemental Information

Supplemental Information 1 Blastocyst

Represented by the ratio of blastocysts to total cells contained in each group

Supplemental Information 2 Cleavage

Cleavage is defined as the development of oocytes to the two- or four-cell stage, and the ratio of the total number of oocytes to the total number of oocytes at the time of titration is the cleavage rate.

Supplemental Information 3 Relative level of EDU

Represents the proliferative capacity of blastocyst cells.

Supplemental Information 4 Relative expression levels of genes representing different functions

Supplemental Information 5 Relative level of GRP78

Fluorescent expression levels of GRP78.

Supplemental Information 6 Relative level of GRP78-protein

Relative expression levels of the target protein GRP78 to β-actin ratio.

Supplemental Information 7 Relative level of GSH

Representing the antioxidant capacity of early embryos.

Supplemental Information 8 Reletive level of JC1

The ratio of red fluorescence to green fluorescence represents the level of membrane potential within mitochondria.

Supplemental Information 9 Relative level of LC3B

Represents the degree of autophagy during early porcine embryonic development.

Supplemental Information 10 Relative level of ROS

Representing the antioxidant capacity of the embryo.

Supplemental Information 11 Relative TUNEL positive nuclei rate

Representing the level of apoptosis during embryonic development.

Supplemental Information 12 The total cell number

Expressed as number of cells per blastocyst.

Supplemental Information 13 Raw images of protein immunoblots

Additional Information and Declarations

Competing Interests

Author Contributions

Data Availability

Lin Wang and Qi-Long Cao are employed by Qingdao Haier Biotechnology Co.

Xiu-Wen Yuan conceived and designed the experiments, performed the experiments, analyzed the data, prepared figures and/or tables, authored or reviewed drafts of the article, and approved the final draft.

Hao Guo conceived and designed the experiments, performed the experiments, analyzed the data, prepared figures and/or tables, and approved the final draft.

ChaoRui Wang conceived and designed the experiments, performed the experiments, analyzed the data, prepared figures and/or tables, and approved the final draft.

HeWei Ji conceived and designed the experiments, prepared figures and/or tables, and approved the final draft.

YongNan Xu conceived and designed the experiments, authored or reviewed drafts of the article, and approved the final draft.

Xue Rui Yao conceived and designed the experiments, authored or reviewed drafts of the article, and approved the final draft.

Lin Wang performed the experiments, prepared figures and/or tables, and approved the final draft.

QiLong Cao performed the experiments, prepared figures and/or tables, and approved the final draft.

Nam-Hyung Kim analyzed the data, authored or reviewed drafts of the article, and approved the final draft.

Ying-Hua Li analyzed the data, authored or reviewed drafts of the article, and approved the final draft.

The following information was supplied regarding data availability:

The raw data is available in the Supplemental Files.

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
