# Peer review of "Fisetin may protect early porcine embryos from oxidative stress by down-regulating GRP78 levels"

_PeerJ, doi:10.7717/peerj.19198_

## Round 0.1 · original submission · Minor Revisions

As you can see, the reviewers raised several important suggestions to improve the manuscript, which you should address.

·

Basic reporting

This manuscript is clearly written in English, with a well-defined background introduction, accurate references, and accessible raw data.

Experimental design

This experiment is based on clear scientific hypotheses and experimental objectives, with a well-designed and reliable framework, comprehensive ethical review, and clearly described experimental methods.

Validity of the findings

The findings of this study are detailed, and the conclusions are clear.

Additional comments

1. Line 52, this study investigates the effects of fisetin on improving porcine embryo quality and its related mechanisms. This represents only a small aspect of the reproductive process; therefore, the term "mammalian reproductive system" is not appropriate here and should be replaced.
2. Line 65-66, GRP78 is one of the primary regulators of the unfolded protein response (UPR). It interacts with endoplasmic reticulum (ER) sensors such as PERK, IRE1, and ATF6 to modulate the stress response and reduce the accumulation of misfolded proteins. Therefore, the sentence “it has been demonstrated that GRP78 levels are elevated when porcine embryos are externally stimulated, which triggers endoplasmic reticulum stress (ERS)” is inaccurate and should be rewritten. GRP78 does not “trigger” endoplasmic reticulum (ER) stress; instead, it acts as its sensing component.
3. Line 127, JC-1 is used to detect mitochondrial membrane potential, but not matrix metalloproteinase.
4. Line 241, This study reported that fisetin improves embryo quality by enhancing mitochondrial function. However, Caspase 8 is a key protein mediated by the death receptor apoptosis pathway. Please explain why was Caspase 8 tested instead of Caspase 9?
5. Line 284-286, “In this study, it has been shown that FIS increases embryos' potential for their mitochondrial membranes.” This sentence is difficult for readers to understand and is not closely related to the following content. It is recommended to rewrite it.
6. Line 291, “oxidative stress”? or “endoplasmic reticulum stress”?
7.Line 325-326, the author is discussing how fisetin improves embryo quality rather than oocyte quality. Therefore, the focus here should be on the role of mitochondria in embryo development.
8. Line 357-358, autophagy is related to oxidative stress, but not all autophagy is induced by ROS. It is recommended to revise the discussion on the mechanisms of autophagy induction.

Reviewer 2 ·

Basic reporting

In this study, the author used porcine oocytes as an experimental model ,in-depth study of the regulatory role of Fisetin on endoplasmic reticulum stress and mitochondrial function during embryo development helps to further clarify the relationship between oxidative stress and embryo development and the regulatory mechanism of related signaling pathways in embryo development, providing valuable theoretical supplements for the fields of reproductive medicine and developmental biology. Nevertheless, there are some points that need to be addressed.
In the part where porcine oocytes and embryos are mentioned as a model for studying the dietary effects on human ovaries, only one reference ([5]) is cited, which is somewhat insufficient. Consider adding relevant research literature to more fully illustrate the similarity between the two and the rationality of this model.

Experimental design

Given that this study solely employed porcine oocytes as the model and all experiments were carried out in vitro, the authors are obliged to discuss the generalizability and limitations of the research findings in the discussion section.

Validity of the findings

The authors could add a column in the existing result to show the BAX/BCL2 ratio (Figure 3C).

Additional comments

There are some grammar errors and inappropriate word usages in the text. For example, in "Fisetin may protects early porcine embryos", "protects" should be changed to "protect".

Reviewer 3 ·

Basic reporting

This report is clear
Figures, those have name of genes must be italic, please revise throughout manuscript.

English in some part difficult to understand, if possible, please edit by professional English editor.

Experimental design

Well design

Validity of the findings

This report showed the novel results of using Fitesin in IVC medium, reduced ROS, apoptosis, ER stress; and increased blastocyst rates, GSH level, expression of pluripotent genes, mitochondrial activities.

These results can be applied in IVP bovine embryos and testing the effects of Fitesin on pregnancy rates after transfer embryo to recipient.

Additional comments

This report has done only in PA porcine embryos. In future should better examine in IVF and cloned porcine embryos.

Annotated reviews are not available for download in order to protect the identity of reviewers who chose to remain anonymous.

---

## Round 0.2 · accepted · Accept

All the reviewers had no concerns and the manuscript is ready for publication.

·

Basic reporting

This manuscript is clearly written in English, with a well-defined background introduction, accurate references, and accessible raw data.

Experimental design

This experiment is based on clear scientific hypotheses and experimental objectives, with a well-designed and reliable framework, comprehensive ethical review, and clearly described experimental methods.

Validity of the findings

The findings of this study are detailed, and the conclusions are clear.

Additional comments

No further comments.

Reviewer 2 ·

Basic reporting

In this study, the author used porcine oocytes as an experimental model, in-depth study of the regulatory role of Fisetin during embryo development, providing valuable theoretical supplements for the fields of reproductive medicine and developmental biology. The manuscript has been significantly improved.

Experimental design

This experiment is based on clear scientific hypotheses and experimental objectives.

Validity of the findings

The findings of this study are detailed, and the conclusions are clear.

Additional comments

The authors have revised the manuscript in accordance with the reviewers' comments, resulting in an overall clearer logic and more distinct data presentation.

Reviewer 3 ·

Basic reporting

Clear and unambitious
Sufficient references

Experimental design

Well design

Validity of the findings

Satisfy all data